# Embryonic Exposure to Bisphenol A Impairs Primordial Germ Cell Migration without Jeopardizing Male Breeding Capacity

**DOI:** 10.3390/biom9080307

**Published:** 2019-07-25

**Authors:** Marta Lombó, Lidia Getino-Álvarez, Alexandra Depincé, Catherine Labbé, María Paz Herráez

**Affiliations:** 1Department of Molecular Biology, Faculty of Biology and Environmental Sciences, Campus de Vegazana, Universidad de León, 24071 León, Spain; 2Fish Physiology and Genomics Department, Campus de Beaulieu, INRA, 35000 Rennes, France

**Keywords:** bisphenol A, primordial germ cells, epigenetics, reproduction, zebrafish

## Abstract

A large amount of chemicals are released to the environment each year. Among them, bisphenol A (BPA) is of utmost concern since it interferes with the reproductive system of wild organisms due to its capacity to bind to hormone receptors. Additionally, BPA epigenotoxic activity is known to affect basic processes during embryonic life. However, its effects on primordial germ cells (PGCs) proliferation and migration, both mechanisms being crucial for gametogenesis, remain unknown. To investigate the effects of BPA on PGCs migration and eventual testicle development, zebrafish embryos were exposed to 100, 2000 and 4000 µg/L BPA during the first 24 h of development. Vasa immunostaining of PGCs revealed that exposure to 2000 and 4000 µg/L BPA impaired their migration to the genital ridge. Two pivotal genes of PGCs migration (*cxcr4b* and *sdf1a*) were highly dysregulated in embryos exposed to these doses, whereas DNA methylation and epigenetic marks in PGCs and their surrounding somatic cells were not altered. Once embryos reached adulthood, the morphometric study of their gonads revealed that, despite the reduced number of PGCs which colonized the genital ridges, normal testicles were developed. Although H3K9ac decreased in the sperm from treated fishes, it did not affect the progeny development.

## 1. Introduction 

Endocrine disruptors (EDCs) comprise a wide variety of substances from different nature whose structure is very similar to that of endogenous hormones [1]. For this reason, EDCs are able to interfere with the synthesis, metabolism, and action of endogenous hormones, endangering human and wildlife health [2]. Bisphenol A (BPA) is one of the most common EDCs produced worldwide, since it is used for the manufacturing of polycarbonate plastics and epoxy resins [3]. Each year, the global production of BPA exceeds 3 million tons and around 3 tons of this compound are released to the environment annually, most of it being leached into the aquatic systems [4]. Concentrations of BPA measured in these ecosystems vary from 21 µg/L in river water samples to 17.2 mg/L in landfill leachates [5]. As a consequence, aquatic organisms are usually exposed to this toxicant, fish populations representing a very vulnerable community [5].

The adverse effects of BPA on reproduction have become one of the major concerns, particularly in the aquatic field, since BPA has been described to alter sex determination, when exposure occurs during gonadal organogenesis [6], and to disrupt gonadal function, when treatment is applied to zebrafish both during embryonic, pubertal [7,8] and adult life [9,10]. Moreover, male exposure to BPA has been demonstrated to reduce the sperm volume, sperm counts, and sperm mobility in goldfish [11], guppies [12], and trout [13]. The effects promoted by paternal BPA treatment during adult life negatively affect the reproductive performance, increasing the percentage of malformations in the progeny outcome [14].

Gonadal development depends on the colonization of the genital ridges by the germ cells during early development. Hu and colleagues showed that exposure of zebrafish embryos to another EDC, 17 α-ethynylestradiol (EE2), impaired primordial germ cell (PGC) migration to the genital ridge [15]. Nonetheless, the impact of BPA exposure during this crucial event on male reproductive performance still remains unknown. PGCs migrate through different tissues from their original extraembryonic location to the genital ridge, where somatic cells involved in gonadal formation are placed [16]. As in many other species, PGC migration in zebrafish can be divided in several steps which starts at dome stage (4.30 h post fertilization, hpf) and ends at 24 hpf [17]. This process is based on a chemotactic gradient in which molecules produced from both PGCs (the receptor Cxcr4b) and somatic cells (the chemokine Sdf1a and the decoy receptor Cxcr7b) are essential [16]. It has been demonstrated that the expression of some genes encoding these molecules, particularly *cxcr4b*, is regulated via estrogen receptors [15,18], so this process might well be a feasible target for EDCs, including BPA.

In this study, we have used zebrafish as model species since the mechanisms underlying PGC migration have been deeply described [16,19,20]. Zebrafish PGCs are characterized by a larger size (10–20 μm) than somatic cells and by the presence of electrodense intracytoplasmic vesicles usually associated with mitochondria, known as *nuage* or germplasm [21]. In the last two decades, a large number of PGC markers have been established, *Vasa* being one of the most useful ones [19,22]. The *vasa* gene encodes an ATP-dependent RNA helicase of the DEAD (Asp-Glu-Ala-Asp) box family [23], whose discovery has enabled scientists to better understand the molecular mechanisms underlying PGC migration [16], to perform PGC transplantation [24], and to generate in vitro PGC-like cells [25].

In addition to the endocrine disruptive effects of BPA, this toxicant has the ability to alter some epigenetic factors in charge of regulating gene expression, thus it has been widely considered as an epigenetic toxicant [9,10,26]. Throughout the life of organisms, the epigenome undergoes two major epigenetic reprogramming waves: during PGC determination and migration and, after fertilization, during the first steps of embryo development [27]. Before gametogenesis, all epigenetic marks in PGCs are erased, allowing the gametes to acquire a new epigenetic pattern during differentiation [28]. The intense epigenetic activity could render these reprogramming waves particularly vulnerable to the effects of BPA.

In this scenario, our hypothesis is that BPA exposure during early embryo development could impair PGCs migration and modify their epigenetic status, having long-term effects on male reproductive performance. To prove this hypothesis, zebrafish embryos were exposed to BPA from fertilization up to 24 hpf. Then, during embryo development, we evaluated PGCs migration, the expression of genes regulating the process (*cxcr4b*, *cxcr7b* and *sdf1a*), and the epigenetic profile of PGCs (DNA methylation and acetylation marks). Once these embryos turned into mature males, the development of the gonads, the epigenetic profile of spermatozoa, as well as the reproductive performance (fertilization success and malformation rate in the progeny) were assessed.

## 2. Results

### 2.1. Germ Cell Migration

In control embryos, at least 12 PGCs managed to migrate to the genital ridge by the first 24 h post fertilization (hpf), whereas this number dropped to around 2 in embryos exposed to 2000 µg/L BPA and to less than 2 on average in embryos exposed to 4000 µg/L BPA from fertilization to 24 hpf (Figure 1A). Most PGCs of embryos treated with these two doses of BPA displayed an ectopic location at 24 h, being positioned between the bud tail and the genital ridge (Figure 1B).

Gene expression of crucial factors for germ cell migration was diversely affected in the genital ridges of embryos exposed to BPA. The expression of *cxcr4b*, a specific gene of germ cells, was repressed in embryos exposed to 4000 μg/L BPA. On the contrary, *sdf1a*, which is expressed by the somatic cells of genital ridge, showed a strong overexpression in embryos treated with 2000 and 4000 μg/L BPA. However, the expression of *cxcr7b* in somatic cells remained steady (Figure 1C).

### 2.2. Epigenetics of Genital Ridges

Global DNA methylation was assessed in order to identify whether epigenetic marks set up during genital ridge formation were affected by BPA. The relative percentage of 5mC in DNA of genital ridges extracted from control embryos showed very low values (8.88% ± 0.97%), displaying a similar profile in embryos exposed to all doses of BPA (Figure 2A).

Although DNA methylation, usually considered as a repressive mark, was not thoroughly affected in genital ridges after BPA exposure, a more accurate analysis of a permissive epigenetic mark (the acetylation of lysine 9 in histone 3 (H3K9ac)) was performed in the PGCs and the somatic cells present in the genital ridges of 24 hpf control and BPA-exposed embryos. Results did not show any difference in H3K9ac levels either in germ cells or in the surrounding somatic cells when comparing control and treated embryos (Figure 2B,C).

### 2.3. Gonadosomatic Index and Morphometric Study in Testicles 

Control embryos and embryos exposed to 4000 μg/L BPA were reared up to the adulthood in order to assess the consequences of the reduction in PGCs showed at earlier stages. When comparing the gonadosomatic index (GSI) from control fish and fish exposed to BPA during embryonic life, no significant differences were observed (Figure 3A). The histological study showed that testicles display a normal structure and that the proportion of testicular cell populations (spermatogonia, spermatocytes, spermatids, and spermatozoa) was not modified by embryonic BPA exposure during the first 24 h of development (Figure 3B,C).

### 2.4. Sperm Epigenetics 

Besides the analysis of the gonads, the sperm of fish exposed to 4000 μg/L BPA during the migration of PGCs was collected to assess the epigenetic profile of spermatozoa. Results showed that BPA treatment during PGCs migration did not affect the hypermethylated profile of zebrafish sperm (81.2% ± 1.03% in control fish and 80.8% ± 1.92% in exposed fish) (Figure 4A). However, a decrease in H3K9ac of spermatozoa from males treated with BPA during PGCs migration was noticed when comparing to sperm obtained from control males (3.5 ± 0.88 in sperm of control fish and 1.68 ± 0.5 in sperm of exposed fish) (Figure 4B,C).

### 2.5. Gonadal Differentiation and Male Fertility

The sex ratio, highly variable among replicates, did not show differences when comparing control fish to those treated with BPA during the first 24 h of development (Table 1). Male fertility was also assessed to prove their breeding capacity. After mating four times, the males of three replicates, the number of fertilized eggs at 3 hpf tended to be higher in spawns fertilized by control fish than in those of exposed fish (Table 1). Although these differences were not statistically significant (*p* > 0.05), an absence of changes between control and exposed group cannot be assumed because the statistical power is pretty low (power < 80%).

In addition to the breeding capacity, paternal contribution was also evaluated by measuring the cumulative embryo mortality and the percentage of larval malformations. Mortality was assessed from 3.3 to 120 hpf and no changes were reported in any stage of development (Figure 5A). Regarding the percentage of malformed larvae, mainly consisting of cardiac and skeletal deformities, no changes were observed when comparing the progeny obtained from control males to that from males exposed to 4000 μg/L BPA during early development (Figure 5B).

## 3. Discussion

Previous studies have already reported that other endocrine disruptors are able to impair PGCs differentiation and migration [15] and they can also affect the sexual determination and breeding capacity of zebrafish [29,30,31]. However, even if it has been already demonstrated that embryonic exposure to BPA jeopardizes embryonic development [32,33], there is little information about the possible effects of BPA on germ cells.

Germ cell migration is a critical period which requires an outstanding communication between PGCs and their somatic microenvironment. This coordination brings about a chemotactic gradient which involves several molecules: the chemokine Sdf1a (also known as Cxcl12a) and its receptor Cxcr4b as well as the decoy receptor Cxcr7b [20]. Taking into account that BPA is considered an endocrine disruptor [2] and that the expression of *cxcr4b* in zebrafish is regulated by estrogen receptors (Esr) [18], alterations in these receptors induced by BPA exposure may cause an imbalance of *cxcr4b*, resulting in PGC migration failure. Indeed, our results showed that exposure to the highest doses (2000 and 4000 µg/L BPA) during the first 24 h of development drastically impaired migration of PGCs, which displayed ectopic locations nearer the tail bud. Two molecular changes might well lie behind the failed migration. The first one implies the downregulation of *cxcr4b* expression observed in PGCs of embryos exposed to 4000 µg/L BPA and the other is related to the overexpression of *sdf1a* in somatic cells of embryos treated with both 2000 and 4000 µg/L BPA, which could prevent this molecule from being removed by Cxcr7b, thus disrupting the chemotactic gradient towards the gonad. Although we have not proved that the impairment of PGCs migration caused by BPA exposure is a result of its activity as endocrine disruptor, Hu and colleagues [15] demonstrated that zebrafish embryonic exposure to EE2 during the first 24 h also led to PGC ectopic locations and, interestingly, this effect was counteracted after Esr2a inhibition, proving that the process of PGC migration depends on estrogenic responses.

The toxicity of BPA has been widely described to affect zebrafish reproduction, in both adult females [9,34] and males [7,10]. Our study presents a different approach, describing the long-term effects of BPA exposure during early embryonic life on male reproduction. As far as testicle formation is concerned, exposure to the highest dose of BPA during the first 24 h of development did not affect the development of the male gonads, neither the GSI. Indeed, histological study revealed a proper gonad structure in males exposed to BPA during PGC migration. This is in contrast to the results obtained from Chen and colleagues [8], who showed lower weight of testicles and sperm count in zebrafish treated with the toxicant from embryonic to adult life: from 8 hpf to 5 mpf (months post fertilization). Other studies have also reported that developmental exposure to other EDCs led to more fibrotic testicles in different fish species [31,35,36]. As reported by Saito and colleagues [37], we have observed that in zebrafish only a dozen of PGCs colonize and generate all germ cells of the gonads in control animals. Nonetheless a complete germ cell replacement can be achieved with a much lower number, as we have noticed in BPA-treated embryos and as it was observed in transplant experiments when one single donor PGC allowed gonadal formation supporting a proper spermatogenesis [24]. This counteracting effect could be explained by the fact that the ability of PGCs to proliferate depends on their number but, once they overtake a threshold, feedback mechanisms start, thus triggering compensatory growth [38]. Our results suggest a rapid PGC proliferation once they reach the genital ridge, a moment in which these cells restart mitosis to produce spermatogonial stem cells [39].

Zebrafish sexual differentiation is environmentally regulated, being susceptible to multiple factors such as stress or rearing density [40], but it is also sensitive to the number of PGCs that colonizes the gonad: Ziwi zebrafish mutants as well as zebrafish *dnd*-morphants, both of them having reduced number of PGCs, tended to male differentiation [41]. Moreover, zebrafish embryos in which transplantation of a single PGC was performed exclusively developed as males [24]. Despite the fact that only a few PGCs reached the genital ridge in embryos exposed to 4000 μg/L BPA, the sex ratio of control and treated fish was essentially the same and so it was the breeding capacity. High doses of BPA, such as the one tested in this study, are supposed to promote feminization, increasing the female ratio in zebrafish [7,42], but the combined interference of estrogenic signaling and PGCs migration could also explain the lack of differences in the sex ratio of BPA-treated fish when compared to the control group. Moreover, the absence of differences could be a result of the small sample size and the high variability among replicates, that prevents us from firmly confirming that sexual differentiation or reproductive capacity were not affected by embryonic BPA exposure.

Epigenetic alterations have been recently described to lie behind the alterations of zebrafish gonad homeostasis [9] and sexual development [40]. In mammals, it has been stated that before gametogenesis takes place, germ cells undergo an epigenetic reprogramming in which they erase most epigenetic marks to delete the epimutations caused during embryonic life or inherited by progenitors [43]. This deletion affects DNA methylation through active or passive mechanisms, so PGCs reached a very hypomethylated status known as “epigenetic ground state” [44]. As for histone marks, there is a general loss of repressive histone modifications (H3K9me2), whereas activating histone modifications (H3K4me2/me3 and H3K9ac) are usually enhanced in PGCs just before germ cell migration in mice [39]. All together these data suggest that exposure to epigenotoxic factors, such as BPA [26], during PGC determination and migration can interfere with their epigenetic reprogramming further affecting gametogenesis. For this reason, both DNA methylation and histone acetylation in genital ridge were analyzed, however, no changes were found either in 5mC or H3K9ac levels when comparing control to treated embryos. The information about the epigenetic pattern of PGCs in fish is very scarce [45]. In fact, there is only one study correlating the DNA methylation in the promoters of *Vasa* and *cxcr4b* in the genital ridge of zebrafish with the expression levels of these two genes [46]. Here we have shown for the first time that the percentage of global DNA methylation in the genital ridge of zebrafish is also very low (around 9%) and that the levels of H3K9ac were similar in PGCs and somatic cells. The global DNA methylation of spermatozoa of males exposed to BPA during embryonic life was not altered either. However, a decrease in acetylation of lysine 9 in H3 was observed in spermatozoa of treated fish. In humans, H3K9ac has been reported to be present in male germ cells and to be involved in spermatozoa development [47]. A decrease in H3K9ac was also described in testicles from rats exposed to a “safe” dose of BPA [48], but in zebrafish, both embryonic and adult exposure increased H3K9ac in blastomeres and testicular cells, respectively [10,33]. Laing and co-workers [9] revealed that exposure of zebrafish breeding couples to BPA led to epigenetic changes that, when affecting expression of genes involved in reproduction, caused a reduction of fertilization rates. Regardless of the alterations in H3K9ac of sperm triggered by embryonic BPA exposure, both male fertility and F1 embryo development were similar to control groups. Even though epigenetic toxicity of BPA in male germline has been demonstrated in the present study, these modifications as well as their ultimate consequences depend on the window of exposure to the toxicant [49]. In this regard, a previous study of our group [10] showed that paternal exposure to lower doses of BPA during the whole spermatogenesis is much more detrimental for male reproduction than embryonic exposure: not only is the proportion of spermatocytes decreased in the testicles of exposed males, but also the fertilization rates obtained are significantly lower. In contrast, the impact of precocious exposure did not promote noticeable long-term effects, being successfully counteracted throughout the development.

## 4. Materials and Methods

### 4.1. Ethics Statement

This work is included in a project from the Spanish Ministry of Economy and Competitiveness (Project AGL2014-53167-C3-3-R) specifically approved by the University of León Bioethical Committee. All the animals were manipulated in accordance with the Guidelines of the European Union Council (86/609/EU, modified by 2010/62/EU), following Spanish regulations (RD 1201/2005, abrogated by RD 53/2013) for the use of laboratory animals.

### 4.2. Zebrafish Maintenance and Embryo Collection

Four-month-old zebrafish (*Danio rerio*), AB strain (wildtype), were maintained in 2.5 L aquaria (ZebTEC, Tecniplast System) with a recirculating water system (pH 7.0–7.5, 30 mg/L Instant Ocean, at 27–29 °C, 14:10 light-dark cycle). Animals were fed twice a day with dry food (Special Diets Services^®^).

To obtain the embryos, adults were mated according to a sex ratio 1:2 (male:female). Embryos were immediately rinsed 2 min in 0.5% (vol/vol) bleach and 10 s in 70% (v/v) ethanol. Next, they were transferred to egg water containing 0.038mM CaCO_3_, 0.446mM NaHCO_3_, 1.025mM sea salt and 0.005% (vol/vol) methylene blue.

### 4.3. BPA Embryonic Exposure

After washing the embryos, 80 of them were transferred for each replicate to a Petri dish containing: egg water with 0.0175% (vol/vol) ethanol (control embryos) or 100, 2000, or 4000 μg/L BPA (exposed embryos). These doses had already been used in zebrafish embryonic exposure: Lam and colleagues [32] had already demonstrated in zebrafish that embryonic exposure to 100 µg/L BPA led to transcriptomic alterations and that concentrations >1500 µg/L BPA significantly increased the percentage of embryo mortality. Moreover, we also proved in a previous study that exposure of zebrafish embryos to 2000 and 4000 μg/L BPA from fertilization to 24 hpf disrupted early development [33]. Around 10 experiments of four replicates per treatment were done to obtain the samples for all the tests. Embryos were incubated from fertilization to 24 hpf, the moment when the PGCs arrive to the genital ridge. From this point onwards, embryos were kept in embryo medium at 28 °C in darkness until further analysis were done.

### 4.4. Whole Mount Immunostaining

For analysis of germ cell migration and germ cell acetylation, the following immunostaining protocol was performed. Control and exposed embryos were collected at 24 hpf and fixed in 4% paraformaldehyde overnight at 4 °C. Permeabilisation with methanol 2 h, at −20 °C was done after washing twice in PBS 1 X and removing the chorions. Then, all embryos were washed 3 times with TBS-T 1% (Tris buffered saline with 1% (vol/vol) Triton X-100) and rinsed in blocking solution (3% (wt/vol) BSA in TBS-T 1%) for 2 h at room temperature. Incubation with both primary antibodies (anti-Vasa, ref.: ab209710; antiH3K9ac, ref.: ab12179) was done for 2 days at 4 °C in blocking solution. Next, they were incubated with fluorescence-conjugated secondary antibodies (goat anti-rabbit AlexaFluor^®^488 and goat anti-mouse AlexaFluor^®^568, respectively) at 4 °C overnight whereas nuclei were stained with 180 μM DAPI (4′,6-diamidino-2-phenylindole) for 8 min. ProLong^®^ Gold Antifade Mountant was added to avoid fluorescence fading whilst using confocal microscope LSM 800 (Zeiss) to observe the embryos placed in ibidi^®^. Intensity of secondary antibodies relative to nuclear areas was established using Fiji ImageJ.

### 4.5. RNA Extraction and qPCR (Quantitative Polymerase Chain Reaction) Analysis

Genital ridges of 20 embryos per replicate were manually dissected from 24 hpf embryos using watchmakers forceps under light microscope. They were transferred to tubes and centrifuged 5 min at 1000× *g*. RNA extraction was done re-suspending the pellets Trizol^®^ reagent according to manufacturer’s protocol. The concentration of RNA samples was measured using the NanoDrop ND-1000 UV-Vis Spectrophotometer, whereas the integrity was assessed by electrophoresis on agarose gel. For cDNA syntheses and RT-qPCR (real time quantitative polymerase chain reaction), the same protocol as those previously described by our group [33] was used. Primer sequences and their annealing temperatures are indicated in Table 2.

### 4.6. Sex Ratio, Fertility, and Early Embryo Development

Control and exposed embryo were maintained, until they reach the sexual maturity, under the conditions mentioned above. At this point, the number of males per 100 females was stablished in each tank. Males were mated with non-treated females and both F1 mortality and larvae malformations were evaluated under stereo microscope (Nikon SMZ1500) throughout the development.

### 4.7. Gonadosomatic Index and Morphometric Study of Testicular Cells

3 groups of 5 control and 5 males exposed to 4000 µg/L BPA were anesthetized with 168 mg/L tricaine (MS222) and weighted. Then, testicles were extracted from each individual and weighted so as to establish the relation of the gonad size to the whole body, measure known as the gonadosomatic index (GSI).

For morphometric study, testicles were rinsed in PBS (8.37mM Na_2_HPO_4_, 1.83mM KH_2_PO_4_, 149.9mM NaCl; pH 7.4) and they were fixed in Bouin solution (75% (vol/vol) picric acid, 7.2% (vol/vol) formaldehyde and 5% (vol/vol) glacial acetic acid). Once gonads were embedded in paraffin, routine histology protocol and hematoxylin-eosin staining was performed. As for morphometric analyses, pictures of 4 µm-slice were taken with a light microscope (Nikon Eclipse E600) and the percentage of every cell type in each testicle was determined with Fiji ImageJ as previously described by our group [10].

### 4.8. Global DNA Methylation in Genital Ridge and Sperm

Genital ridges were extracted from 20 control and treated embryos at 24 hpf, whereas sperm pools were obtained by abdominal massage of 5 males per replicate (3 replicates per treatment) to get around 10^8^ cells/mL. 680 μL TNES buffer (125 mM NaCl, 10 mM EDTA, 17 mM SDS, 4 M urea, 10 mM Tris-HCl, pH 8) were added to each sample plus 20 μL of proteinase K 200 μg/mL and they were incubated overnight at 56 °C. DNA was precipitated following the instructions of Marandel and colleagues [50], with some modifications. DNA concentration and purity were measured using the NanoDrop ND-1000 UV-Vis to ensure high purity (A260/A280 > 1.8).

Analysis of global methylation (referred as relative amount of 5 methylcytosine, 5mC) in genital ridges was carried out by UPLC-MS (Ultra-performance liquid chromatography - mass spectrometry) following the protocol described by Le and colleagues [51]. Sperm global DNA methylation was assessed by LUMA (Luminometric Methylation Assay) using a pyrosequencer (PyroMark Q96 ID). Three pools of sperm DNA (600 ng) per treatment were digested with both MspI/EcoRI and HpaII/EcoRI and products were pyrosequenced. In order to calculate the percentage of DNA methylation, the HpaII/EcoRI ratios were obtained as described by Karimi and colleagues [52].

### 4.9. Immunostaining of Epigenetic Marks in Spermatozoa

3 pools of sperm from 5 control and 5 males exposed to 4000 µg/L BPA during embryonic life were fixed with 4% (wt/vol) paraformaldehyde for 20 min at room temperature. After washing the samples twice with bi-distilled water, 20 µL of spermatozoa from each individual male (3 per treatment) were spread out on ATE ((3-aminopropyl)trimethoxysilane) coated slides at 37 °C overnight. The following steps were carried out as indicated in our previous work [53]. The primary antibody anti-H3K9ac was purchased from Cell Signaling (ref.: C5B11). Intensity of secondary antibody relative to nuclear areas was established using Fiji ImageJ.

### 4.10. Statistical Analyses

All of the tests were done with SPSS Statistics 24.0 (IBM, USA). First of all, normality of the data was checked using the Shapiro-Wilk test. Concerning embryonic exposure, parametric data were analyzed using one-way ANOVA followed by Bonferroni post hoc test, whereas for non-parametric data a Kruskal–Wallis test was applied. Regarding the analyses performed in the adulthood, for parametric data a Student’s t-distribution (*p* < 0.05) was used, whereas non-parametric were evaluated with Mann–Whitney–Wilcoxon test (<0.05). Moreover, an analysis of the statistical power was performed in the data with high variability among replicates (sex ratio and number of fertilized eggs) in order to avoid an erroneous acceptance of the null hypothesis.

## 5. Conclusions

In this study we have demonstrated that exposure to 2000 and 4000 µg/L BPA during early embryonic development reduces the number of PGCs in the genital ridge. This decrease might well be a result of the ectopic location of PGCs and the dysregulation of key factors in the process of migration, such as *cxcr4b* and *sdf1a*. Despite the fact that only very few cells reached the genital ridge in embryos treated with 4000 µg/L BPA, they are able to proliferate leading to a normal testicle formation. Regarding epigenetics, even though no changes were observed in germ or somatic cells within the genital ridge, males exposed to BPA during embryonic life exhibited a decrease in H3K9ac of sperm. However, precocious exposure to BPA did not have long-term consequences on male reproductive performance, although a greater number of replicates will be necessary to undoubtedly discard any effect on sexual differentiation or fertility.

## Figures and Tables

**Figure 1 biomolecules-09-00307-f001:**
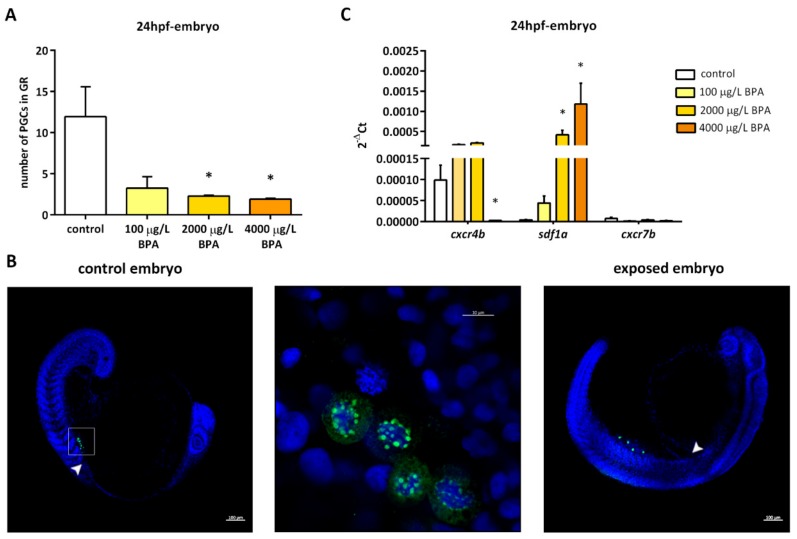
(**A**) Number of primordial germ cells (PGCs) observed in the genital ridge of embryos at 24 h post fertilization (hpf). Bars represent mean of PGCs in 5 embryos at 24 hpf in 3 different experiments (*n* = 3). Asterisks indicate significant differences (*p* < 0.05) when compared to control groups. (**B**) Location of PGCs (green fluorescence) by confocal imaging in 24 hpf embryos. In control embryos (left picture), they were located within the genital ridge (arrowhead); scale bar 100 µm. The inset region is enlarged in the middle picture, in which germ cells are marked with Vasa (green spots surrounding nuclear area stained with DAPI); scale bar 10 µm. In 24 hpf embryos exposed to 4000 µg/L BPA (right picture), PGCs appeared outside the genital ridge (arrowhead); scale bar 100 µm. (**C**) Relative expression of genes involved in primordial germ cell migration in the genital ridges of control and BPA-exposed embryos. Expression levels relative to *18 S rRNA* were calculated using 2^−ΔCt^ method in three independent experiments (*n* = 3). Asterisks indicate significant differences (*p* < 0.05) when compared to control embryos.

**Figure 2 biomolecules-09-00307-f002:**
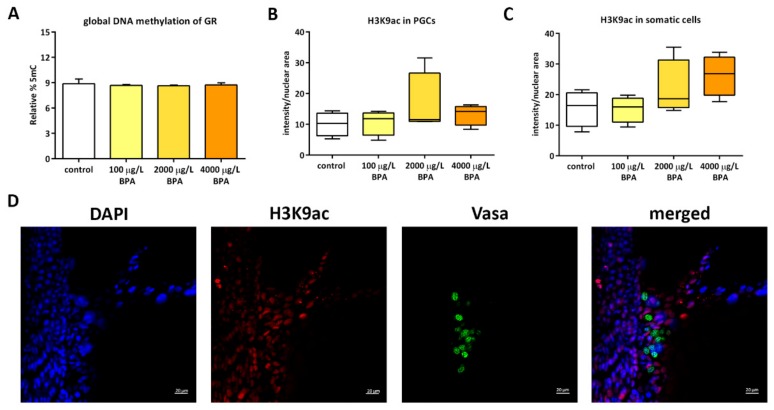
Epigenetic profile of PGCs and their somatic surrounding cells. (**A**) Quantification of 5 mC by UPLC-MS in DNA of genital ridges (GR) from control and BPA-treated embryos. Bars represent the percentage of 5mC relative to C in 3 pools of 20 genital ridges per treatment (*n* = 3). Quantification of H3K9ac by whole mount immunofluorescence in (**B**) germ cells and (**C**) somatic cells of 24 hpf control and BPA-exposed embryos. Boxes represent the median plus maximum and minimum of H3K9ac intensity relative to nuclear area in 4 independent experiments (*n* = 4). (**D**) confocal images of double immunostaining in genital ridges of 24 hpf control embryos. H3K9ac was labelled with AlexaFluor^®^568 (red fluorescence), Vasa protein within PGCs was labelled with AlexaFluor^®^488 (green fluorescence) and nuclei were stained with DAPI (in blue); scale bar 20 µm.

**Figure 3 biomolecules-09-00307-f003:**
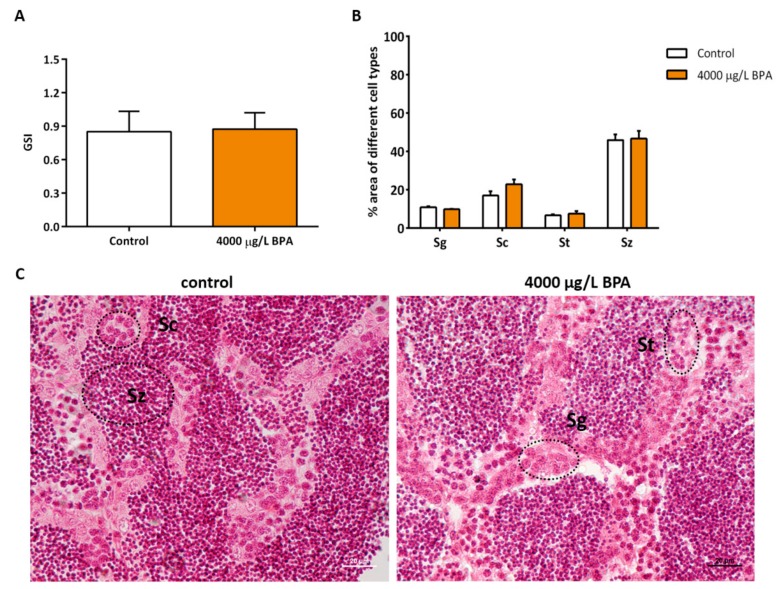
Evaluation of testicular status in adult males. (**A**) GSI (gonado-somatic index) of testicles from 5 control and 5 BPA-exposed fish in each of the 3 different replicates (*n* = 3). (**B**) morphometric study showing the percentage of each cell type: sg (spermatogonia), sc (spermatocyte), st (spermatid), and sz (spermatozoa) of both testicles from 3 different males per treatment (*n* = 3). (**C**) representative images of testicular cell types in control and treated fish. 4 µm-slices were stained with hematoxylin-eosin; scale bar 20 µm.

**Figure 4 biomolecules-09-00307-f004:**
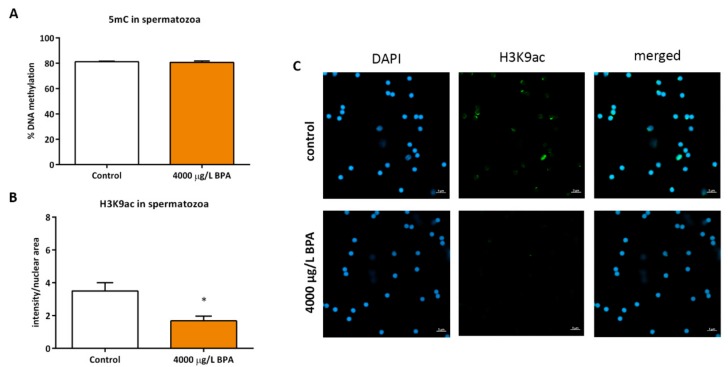
Epigenetic profile of spermatozoa. (**A**) bars represent the percentage of global DNA methylation in spermatozoa of 3 pools of sperm from 5 control and 5 exposed fish each (*n* = 3). (**B**) bars represent the intensity of H3K9ac labelled with AlexaFluor^®^488 of 3 pools of sperm from 5 control and 5 exposed fish (*n* = 3); asterisks indicate significant differences (*p* < 0.05) when compared to control males. (**C**) confocal images of H3K9ac in spermatic nuclei (stained with DAPI); scale bar 5 µm.

**Figure 5 biomolecules-09-00307-f005:**
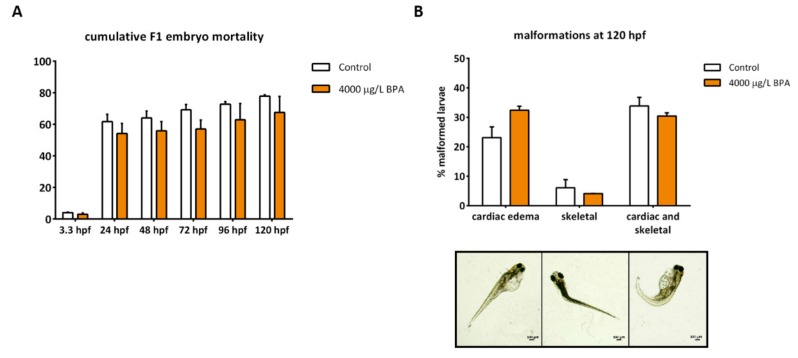
Evaluation of F1 embryonic development. (**A**) cumulative percentage of embryo mortality from 3.3 to 120 hpf of six independent matings per treatment (*n* = 6). (**B**) percentage of malformed larvae at 120 hpf of six independent batches per treatment (*n* = 6). No significant differences between treatments (*p* < 0.05) were found. Images represent the main type of malformations: cardiac edema, skeletal and altogether; scale bar 100 µm.

**Table 1 biomolecules-09-00307-t001:** Sex ratio and male breeding capacity. Female/male rates in three replicates of control and treated fish are shown in left column (*p* = 0.868; power = 5.2%), whereas male fertility indicated as mean of fertilized eggs at 3 hpf in four different spawns per replicate are represented in the right column (*p* = 0.775; power = 5.7%).

		Sex Ratio	Male Fertility (Fertilized Eggs ± SE)
		(Males/Females) × 100	
**Control**	Replicate 1	161.90	429.75
	Replicate 2	47.83	442.77
	Replicate 3	253.33	82.62
	Mean		351.71 ± 146.58
**4000 µg/L BPA**	Replicate 1	189.47	282.33
	Replicate 2	23.81	274.83
	Replicate 3	205.56	289.83
	Mean		282.33 ± 7.5

**Table 2 biomolecules-09-00307-t002:** List of primers designed for gene expression analysis. Sequences start from 5′ to 3′. Amplicon size indicated as number of base pair (bp) and annealing temperature as grade Celsius (°C).

Transcript Name	Primer Set	Amplicon Size	Annealing Temperature	Accession Number
**sdf1a**	F: ATTCGCGAGCTCAAGTTCCT R: ATATCTGTGACGGTGGGCTG	214	62	NM_178307.2
**cxcr4b**	F: GGCGCTGGCATATTTCCA R: ACGCCTAGGAAAGCATAAAGGA	56	60	AY057094.1
**cxcr7b**	F: CGCCAGCATCTTCTTCCTGA R: GCGAATAAAGCAAGCAGCCA	138	64	EF467375.1
**18S rRNA**	F: GCCGTTCTTAGTTGTGGAG R: CCGGAGTCTCGTTCGTTATC	60	58	FJ915075.1

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
