# Peer review of "Embryonic Exposure to Bisphenol A Impairs Primordial Germ Cell Migration without Jeopardizing Male Breeding Capacity"

_biomolecules, 2019, doi:10.3390/biom9080307_

Round 1

Reviewer 1 Report

In this study, Lombó and coworkers addressed the issue of organism exposure to BPA during early embryonic development. They found that the exposure of zebrafish embryos to 4000 µg/L BPA for 24 hours from fertilization reduces the number of primordial germ cells (PGCs) successfully migrating in the genital ridge. However, subsequent testicle formation was unaffected, such as male breeding capacity, leading to the conclusion that the early exposure to BPA do not have long-term consequences on male reproductive performance. The work reports also some genomic details regarding the dysregulation of two genes playing a fundamental role in PGC migration (cxcr4b and sdf1a), together with some epigenomic data about DNA methylation and epigenetic marks in PGCs and genital ridges somatic cells.

Overall, the investigation is interesting and well conducted, by both a methodological and a conceptual point of view. However, a couple of major issues prevent me from suggesting a direct acceptance of the manuscript for publication in Biomolecules journal.

1 - The scope of the work is somehow limited by the use of a single (and rather high) BPA concentration. Due to the strong (and sometimes non-monotonic) dose-dependency of the biological effects triggered by BPA exposure, the use of a single BPA dose can be misleading. The entire work would greatly benefit from the parallel test of multiple BPA doses. At least, Authors should test one additional—and more environmentally relevant—dose (e.g. 1000 or 400 µg/L) for the first assay of PGC migration (and expression of the three reported genes) where a substantial and statistically significant effect has been actually observed.

2 - In this study a relatively large number of negative results are presented. Despite the well-known bias dominating the scientific literature about this type of data report, I am firmly convinced that negative results are informative as much as positive ones (especially when it comes to toxicity issues) and I really appreciated Authors' choice. However, if p-values can be used as a tool to reject a null hypothesis (H0), it should be noticed that, taken alone, p-value is not a sufficient criterion to claim the absence of an effect when failing in rejecting H0. P-value computation do not protect indeed from type II errors (i.e. false negatives). In this sense, male breeding capacity assessment (Table 1) could be an emblematic case. Specifically, computation of statistical power is what it would take. Actually, achieved statistical power can be computed only if the effect size is known, that it is rarely the case; if you suppose (or claim) that there is no effect in your comparison, a statistical sensitivity analysis (with a fixed power of at least 80%) should be conducted. Sensitivity analysis will allow to find the minimum detectable effect size in a particular experimental condition (given alpha=0.05, power=0.8 and sample size=n). Consult a statistician if needed and, please, add this useful information for each comparison, in figure captions.

Afterward, consider to increase your simple size (n) if you realize to have an unsatisfactory statistical sensitivity in some of your comparisons.

Minor Comments

1 – In Figure 3A, there is no information about the number of independent experiment performed. Please add it.

2 – When reporting a percentage along with its uncertainty, e.g. (8.73% ± 0.44), a better notation should be the following: (8.73 ± 0.44)%.

3 - Why there is always an 'I' character after 'Figure X.' in every figure caption? Is it a typo?

Author Response

Major Comments:

1.       We agree with the fact that BPA often triggers biological effects which are dependent of the doses. Therefore, we have also tested the effects of 100 and 2000 µg/L BPA on PGC migration and on the epigenetic status of genital ridges and PGCs. Results showed that the exposure to 2000 µg/L BPA also impaired PGC migration (the number of PGCs that reached the gonadal ridges dropped to around 2) and it led to an overexpression of sdf1a. However, no changes were reported either in the methylation of genital ridges or in the acetylation of PGCs and their surrounding somatic cells.

2.       First of all, reviewer#1 described as “negative results” the lack of significant changes between control and BPA-exposed group. However, the absence of alterations in the development of gonads observed in the males treated with BPA during embryonic development is of great value for us. It reinforces the idea, previously proposed by other authors [1,2], that this species has a great capacity to overcome a failure in development, such as the reduction of PGCs in the gonadal primordium.

The reviewer also claimed that, effectively, P-value computation do not protect from type II errors and ask for a statistical power of 80% in order to avoid failing in rejecting H0. In epidemiological studies, a sample size similar to that used in this work is considered very low. However, when using animals as experimental models the number of exposed or sacrificed individuals must be reduced. Even though here we have often a sample size of 3 (n=3), samples frequently come from a pool of several embryos (20) or males (5), so as to obtain a minimum quantity of material and also to minimise the individual factor.

The statistical power depends on the effect size, the variability of the samples, the sample size and also on the P-value. As a result of the high variability in the sex ratio and the number of fertilised eggs, both of them presented in table 1, and also because of the high value of P-values obtained (p=0.868 and p=0.775, respectively), the statistical power is very low (power=5.2% and power=5.7%, respectively). Therefore, we cannot conclude, as the reviewer suggested, that exposure to BPA during embryonic life has none effect on these two parameters and we have changed the discussion and conclusions according to these results. We have also explained in the discussion that, in zebrafish, sex ratio depends on multiple factors, being density one of them [3]. So the high variability in this parameter might be biased by the different number of fishes in each tank.

Nevertheless, an increase in the sample size would require at least 6 months to allow exposed fishes to reach the adulthood and, what’s more, we would have to expose great many embryos since only few of them are able to survive to mature period, when most of the parameters (sperm epigenetic status, gonadal development, sexual differentiation and progeny performance) have been studied. Moreover, each analyses requires the use of several fishes. For instance, to obtain each sample of sperm we have pooled samples of at least 5 fishes per replicate, since each individual male only produces a maximum of 2 µL of sperm.

Minor Comments:

1.       It has been added. For the analysis of GSI we used 5 males per replicate and a total of 3 replicates per treatment (n=3)

2.       Ok.

3.       It is just a bar to separate the figure from the text. It has been removed to avoid any confusion.

Reviewer 2 Report

Overall it is a good article with sound science. My questions are:

In the paper, the selection of dose needs to be justified (as 4000 ug/L BPA is higher than human physiological dose)

The use of sample size (n = 3) in figure 4 might be small to draw a valid statistical conclusion.

Author Response

Minor Comments:

1.         The doses have been selected based on other studies in this species. These doses had already been used in zebrafish embryonic exposure: Lam and colleagues [4] had already demonstrated in zebrafish that embryonic exposure to 100 µg/L BPA led to transcriptomic alterations and that concentrations >1500 µg/L BPA significantly increased the percentage of embryo mortality. Moreover, we also proved in a previous study that exposure of zebrafish embryos to 2000 and 4000 μg/L BPA from fertilization to 24 hpf disrupted early development [5]. Indeed, human exposure implies very long periods and lower doses of BPA. Here, we have used some doses included within those determined in aquatic systems, so as to evaluate the risks for fish populations [6]. The rationale for the use of these doses has been included in M&M.

2.         Although the n in figure 4 is 3, each sperm sample has been obtained from 5 males per replicate to avoid the “individual factor”. Moreover, in each of these three replicates we have analyzed around 200 spermatozoa, so we considered that the cell population we have studied is quite representative. We have provided this information in M&M and in the figure legends.

References

1.          Saito, T.; Goto-Kazeto, R.; Arai, K.; Yamaha, E. Xenogenesis in Teleost Fish Through Generation of Germ-Line Chimeras by Single Primordial Germ Cell Transplantation1. Biol. Reprod. 2007, 78, 159–166.

2.          Tzung, K.W.; Goto, R.; Saju, J.M.; Sreenivasan, R.; Saito, T.; Arai, K.; Yamaha, E.; Hossain, M.S.; Calvert, M.E.K.; Orbán, L. Early depletion of primordial germ cells in zebrafish promotes testis formation. Stem Cell Reports 2015, 4, 61–73.

3.          Ribas, L.; Vanezis, K.; Imués, M.A.; Piferrer, F. Treatment with a DNA methyltransferase inhibitor feminizes zebrafish and induces long-term expression changes in the gonads. Epigenetics Chromatin 2017, 10, 59.

4.          Lam, S.H.; Hlaing, M.M.; Zhang, X.; Yan, C.; Duan, Z.; Zhu, L.; Ung, C.Y.; Mathavan, S.; Ong, C.N.; Gong, Z. Toxicogenomic and phenotypic analyses of bisphenol-a early-life exposure toxicity in zebrafish. PLoS One 2011.

5.          Lombó, M.; González-Rojo, S.; Fernández-Díez, C.; Herráez, M.P. Cardiogenesis impairment promoted by bisphenol A exposure is successfully counteracted by epigallocatechin gallate. Environ. Pollut. 2019, 246, 1008–1019.

6.          Crain, D.A.; Eriksen, M.; Iguchi, T.; Jobling, S.; Laufer, H.; LeBlanc, G.A.; Guillette, L.J. An ecological assessment of bisphenol-A: Evidence from comparative biology. Reprod. Toxicol. 2007, 24, 225–239.

Round 2

Reviewer 1 Report

 The authors have satisfactorily addressed all my concerns and made the necessary changes to the manuscript.

Some very minor typos remain to be corrected:

line 89 in Section 2.1:

"exposed to with 4000 µg/L BPA" should be "exposed to 4000 µg/L BPA"

In legend of Figure 5:

"mattings" should be "matings"

in line 243 of Discussion Section:

"result of the low number of sample size" should be either "result of the small sample size" or "a result of the low number of observations".

line 244 of Discussion Section:

"prevents us from firmly confirm that" should be "prevents us from firmly confirming that"

Apart from these few grammar misspellings, I recommend the publication of the manuscript in the present form.

Author Response

Taking into account the comments of reviewer 1, we have accepted in the manuscript our previous changes and, therefore, the corrections that still needed to be addressed have changed their location in the document:

-"exposed to with 4000 µg/L BPA" should be "exposed to 4000 µg/L BPA" is now in line 87

-"result of the low number of sample size" should be either "result of the small sample size" or "a result of the low number of observations". is now in line 239

-"prevents us from firmly confirm that" should be "prevents us from firmly confirming that" is now in line 240.

We'd like to thank also reviewer 1 for making us improve the quality of the present work and for recommending the publication of our manuscript in Biomolecules.